# Cleaning up the air: Effectiveness of air quality policy for $SO_2$ and $NO_x$ emissions in China

Ronald J. van der A[1,2], Bas Mijling[1], Jieying Ding[1,3], Maria Elissavet Koukouli[4], Fei Liu[1], Qing Li[5], Huiqin Mao[5], Nicolas Theys[6]

[1]Royal Netherlands Meteorological Institute (KNMI), De Bilt, The Netherlands
[2]Nanjing University of Information Science and Technology, Nanjing, P.R.China
[3]Delft University of Technology, Delft, the Netherlands
[4]Laboratory of Atmospheric Physics, Aristotle University of Thessaloniki, Thessaloniki, Greece
[5]Satellite Environment Center, Ministry of Environmental Protection, Beijing, P.R.China
[6]Belgian Institute for Space Aeronomy (BIRA-IASB), Brussels, Belgium

*Correspondence to*: Ronald J. van der A (avander@knmi.nl)

**Abstract.** Air quality observations by satellite instruments are global and have a regular temporal resolution, which make them very useful in studying long-term trends in atmospheric species. To monitor air quality trends in China for the period 2005-2015 we derive $SO_2$ columns and $NO_x$ emissions on a provincial level with improved accuracy. To put these trends into perspective they are compared with public data on energy consumption and the environmental policies of China. We distinguish the effect of air quality regulations from economic growth by comparing them relatively to fossil fuel consumption. Pollutant levels, per unit of fossil fuel, are used to assess the effectiveness of air quality regulations. We note that the desulphurisation regulations enforced in 2005-2006 only had a significant effect in the years 2008-2009 when a much stricter control of the actual use of the installations began. For national $NO_x$ emissions a distinct decreasing trend is only visible since 2012, but the emission peak year differs from province to province. Unlike $SO_2$, emissions of $NO_x$ are highly related to traffic. Furthermore, regulations for $NO_x$ emissions are partly decided on a provincial level. The last three years show both a reduction in $SO_2$ and $NO_x$ emissions per fossil fuel unit, since the authorities have implemented several new environmental regulations. Despite an increasing fossil fuel consumption and a growing transport sector, the effects of air quality policy in China are clearly visible. Without the air quality regulations the concentration of $SO_2$ would be about 2.5 times higher and the $NO_2$ concentrations would be at least 25% higher than they are today in China.

## 1. Introduction

Satellite instruments can monitor air quality from space by mapping e.g. aerosols and tropospheric ozone, but are especially useful for emission estimates in observing the relatively short-living gases nitrogen dioxide ($NO_2$) and sulphur dioxide ($SO_2$). For these two trace gases improved data sets recently became available, enabling analysis of air quality time series on a national or provincial level with improved accuracy. Theys et al. (2015) presented a new data set of $SO_2$ column densities derived from the Ozone Monitoring Instrument (OMI) satellite instrument (Levelt et al., 2006). They conclude that the $SO_2$ concentrations derived from OMI agree on average within 12% with ground observations. This data set strongly improves on earlier $SO_2$ data sets from satellites, which motivated this study. For $NO_2$, instead of using concentration data, we assess directly the emission data of nitrogen oxides ($NO_x = NO_2 + NO$) that was derived from satellite observations by Mijling and Van der A (2012) and removes the meteorological influences. The precision of the derived $NO_x$ emissions per grid cell of 0.25 x 0.25 degree is estimated as 20% (Ding et al., 2016).

China is one of the biggest emitters of $SO_2$ and $NO_2$ into the atmosphere because its large economy depends heavily on fossil fuels as an energy source. China alone is responsible for about 30 % of the global total emissions of $SO_2$ into the atmosphere (Klimont et al., 2013), while over 90% of the $SO_2$ emissions are caused by coal consumption in China (Chen and Xu, 2010).

Coal is mainly used by thermal power plants and energy-intensive industry (e.g. steel, cement and glass), and to a lesser extent by residential use. $SO_2$ is also released by the use of oil and natural gas, but the sulphur content in these fuel types is much lower. Of these sources power plants are responsible for about 30-40 % of all emissions and industry for another 50-60 % (He K. et al., 2012, ChinaFAQs project, 2012). According to the Multi-resolution Emission Inventory for China (MEIC)

(http://www.meicmodel.org/) the source of $SO_2$ emissions in 2010 was 29.4 % from power plants, 57.7 % from industry and 11.7 % residential and 1.2 % from transport. Figure 1 shows the location of the 600 biggest thermal power plants on the map of China including a list of provinces mentioned in this study. At a global scale, volcanic activity is another important source of atmospheric $SO_2$. However, plumes of active volcanoes are seldom observed over China.

$NO_x$ is released by more or less the same anthropogenic sources, i.e. the burning of coal or oil. The main difference with $SO_2$

is that traffic is a much more important source for $NO_x$. $NO_x$ emission factors (i.e. emissions per fossil fuel unit) in the transport sector are generally much higher than emission factors in energy and industry, which makes traffic one of the major sources of $NO_x$ in China. According the MEIC inventory, 25% of $NO_2$ in 2010 was released by traffic, 32% by power plants, 4% by residential sources and 39% by industry, with the cement industry being the biggest emitter in this sector.

To reduce $SO_2$ in China, the authorities have implemented several environmental regulations. The most important regulation

was the desulphurization of coal-fired power plants in 2005/2006 (Xu, 2011). This was later followed in the 12[th] five-year plan (2011-2015) by stricter control on the implementation of the regulations, additional filtering efforts, switching to low-sulphur coal and gasoline, phasing out obsolete capacity in coal-using industry, phasing out of small-scale coal mining, and gradually using more oil, gas and renewable energies instead of coal since 2011. An overview of all regulations related to $SO_2$ is shown in Table 1, which includes the year of the beginning of the implementation.

The regulation of $NO_x$ was started much later than for $SO_2$. The 12[th] five-year plan mentioned the intention to reduce $NO_2$ by 10 % (target) (ChinaFAQs project, 2012), from 2011 onward $NO_x$ filtering systems were installed, mainly at power plants but also for heavy industry. These regulations for $NO_x$ were announced in 2013 in the Air Pollution Prevention and Control Action Plan (CAAC, 2013). According to Liu et al. (2016) Selective Catalytic Reduction (SCR) equipment was installed in this period and growing from a penetration of about 18% in 2011 until 86% in 2015. SCR equipment in power plants are

expected to reduce the emissions of the power plant with at least 70% (ICAC, 2009). The SCR installation is the most significant measure taken to reduce the $NO_x$ emissions and it largely coincides with the peak year of observed $NO_2$ concentrations (Liu et al., 2016). At the same time China has implemented several new national emission standards for cars during the time period of our study (see Table 2). The change from China 3 to China 4 standard for cars in the period 2011-2015 reduces the maximum allowed amount of $NO_x$ emissions for on-road vehicles with 50% (Wu et al., 2017). More strict

regulations for on-road vehicles (e.g. ban on older polluting cars) have been introduced on a city level, e.g. in Beijing, rather than nationwide. To our knowledge no regulations for ship emissions have been announced. Strong regulations have also been enforced during specific events like the Olympic Games in 2008, Shanghai World Expo in 2010, Nanjing Youth Olympic Games in 2014, and the APEC meeting in 2014, but those regulations were mostly of a temporary nature as shown by e.g. Mijling et al. (2012) for the Olympic Games in 2008.

To study the efficiency of the environmental policies, we analysed satellite observations of $SO_2$ and tropospheric $NO_2$ of the last 11 years. $SO_2$ satellite observations over China have been studied earlier by Lee et al. (2010), Li et al (2011), He (2012), Yang et al. (2013), Fioletov et al. (2015), and Krotkov et al. (2015). Satellite observations are very useful for $SO_2$ trend studies, as recently McLinden et al. (2016) showed that bottom-up inventories are underestimating $SO_2$ emissions worldwide with about 0-10%. $NO_2$ satellite observations over China have been evaluated by e.g. Richter et al. (2005), van der A et al.

(2006), Zhang et al. (2012), and Krotkov et al. (2015). All these studies showed a strong increase in $NO_2$ over East China. On a city-scale or regional level trends are analysed and reported by Gu et al.( 2013), Schneider et al. (2015), and Duncan et al. (2016). Although some cities showed already a decreasing trend, notably in the Pearl river delta,. an overall decrease in $NO_2$ concentrations in China is only recently observed by Irie et al. (2016) Liu et al., (2016b) and De Foy et al. (2016). To

exclude meteorology as a factor for variability in $NO_2$ several authors evaluated $NO_x$ emissions instead. Emission estimates of $NO_x$ over China have been analysed by Stavrakou et al. (2008), Kurokawa et al. (2009), and more recently by Mijling et al. (2013) and by Liu et al. (2016a).

In these studies, whether of concentrations or emissions, linear trends of the air pollutants are often used. Here, however, we will relate changes derived on a provincial level for China with the energy consumption and the environmental policies of the country. This gives insight in the efficiency of the applied air quality policies and regulations. We apply this to $NO_x$ emissions instead of concentrations for the period 2007 until 2015. The comparison of $SO_2$ trends with those of $NO_x$ emissions enables us to distinguish environmental policies specifically applied on coal-based industry and power plants with general environmental measures and trends in traffic.

## 2    Observational data

### 2.1 Satellite observations of $SO_2$

$SO_2$ is observed in the UV spectral range of satellite observations of SCIAMACHY (on Envisat), GOME-2 (on METOP-A) and OMI (on EOS-AURA). $SO_2$ retrieval algorithms have been earlier developed for GOME-1 by Eisinger and Burrows (1998), for SCIAMACHY (Lee et al., 2008), GOME-2 and for OMI by Krotkov et al (2006). Recently a new retrieval algorithm has been developed (Theys et al., 2015) that improves the precision of the $SO_2$ data for OMI with a factor 2, allowing us to derive more accurate trends based on OMI. The retrieval method is based on a Differential Optical Absorption Spectroscopy (DOAS) scheme to determine the slant columns from measured spectra in the 312-326 nm spectral range, which are then background corrected and converted to vertical columns using an Air Mass Factor (AMF). The AMF is calculated with the radiative transfer model LIDORT (LInearized Discrete Ordinate Radiative Transfer model). More details about the retrieval procedure are described in Theys et al. (2015). Also the operational algorithm of NASA for $SO_2$ from OMI has recently been improved. This algorithm and the algorithm of Theys et al. (2015) have a very comparable performance as shown by Fioletov et al. (2016).For this study, the algorithm of Theys et al. (2015) has been applied on the observations of the OMI instrument (Levelt et al., 2006) for its whole mission from 2004 onwards

To improve the quality of the OMI $SO_2$ data we exclude observations with a cloud fraction of more than 50 percent or with a fitting chi-square higher than 1. The solar zenith angle is limited to 75° and the viewing angle to 50°. Since the OMI instrument is suffering from the so-called row anomaly since 2007 (KNMI, 2012), we filter the affected rows (24-49, 54-55) in the same way for all years in the time series.

As we focus on anthropogenic $SO_2$, the $SO_2$ data for 15 June - 9 July 2011 have been removed because of its contamination with volcanic $SO_2$ from the eruption of the Nabro volcano in Africa and the transport of its plume to China (Brenot et al., 2014).

As a first step in our study we have made monthly means for the whole data set by averaging and gridding the data to a resolution of 1/8° by 1/8°. The gridding algorithm takes into account the area of each satellite footprint overlapping the grid cell. The resulting data set is a time series of monthly means for the time period October 2004 to December 2015.

For comparison we also use the official ESA SCIAMACHY/Envisat $SO_2$ product version SGP 5.02 and the standard data product from the GOME-2/METOP-A version GDP 4.7, as developed within the EUMETSAT Satellite Application Facility for Atmospheric Composition and UV radiation, O3MSAF, project and distributed by http://atmos.caf.dlr.de/gome2/. The data of these instruments are noisier than the OMI datasets because of the lower spatial coverage, different fit window and the lower signal-to-noise ratio of the SCIAMACHY and GOME-2 instruments. Therefore, their quality-controlled monthly mean $SO_2$ data have been recalculated by spatially averaging for each grid cell the data from the eight surrounding neighbouring cells, hence creating a smoothed $SO_2$ field. For details on the methodology and findings refer to Koukouli et al. (2016).

## 2.2 NO$_x$ emission estimates from satellite observations

For NO$_x$ emission data we use the results of an update (version 4) of the DECSO (Daily Emission estimates Constrained by Satellite Observation) algorithm developed by Mijling and van der A (2012). DECSO calculates emissions by applying a Kalman filter for the inversion of satellite data and a regional Chemical Transport Model (CTM) for the forward model calculation. It takes transport from the source into account with a semi-Lagrangian approach. The CTM we use is CHIMERE v2013 (Menut et al., 2013) with meteorological information from the European Centre for Medium-range Weather Forecasts (ECMWF) with a horizontal resolution of approximately 25x25 km$^2$. The DECSO algorithm is applied to OMI NO$_2$ observations derived by the DOMINO v.2 algorithm (Boersma et al., 2011). The latest improvements of the DECSO algorithm resulting in version 4 are described by Ding et al. (2015, 2016). The monthly average emission data over China we use is available on 0.25 degree resolution for the period 2007-2015 on the web-portal www.globemission.eu.

## 3  Temporal analysis over China

### 3.1 Sources of SO$_2$ and NO$_x$ in China

The multi-annual mean of SO$_2$ for 2005-2015 is shown in Figure 2. As the lifetime of SO$_2$ is relatively short (typically 4-48 hours) (Lee et al., 2011, Fioletov et al., 2015), the observed SO$_2$ concentrations are a good proxy for the location of SO$_2$ emissions. Regions with large SO$_2$ concentrations are South Hebei, the province Shandong (around the city Zibo) and the region around Chongqing. South Hebei is a region with many power plants just east of the mountainous coal-mining area in Shanxi. The hot spot in the province Shandong is related to a strongly industrialized area with lots of coal-using industry. In the Chongqing region both coal mines and heavy industry are located.

Rather than located at hot spots, high NO$_2$ concentrations are more distributed over the East of China, mainly because traffic is an important source of NO$_x$ emissions (see Figure 3a). The underlying NO$_x$ emissions are shown in Figure 3b. Like for the SO$_2$ concentrations, NO$_x$ emission spots can be found at the location of big power plants. Also clearly visible are the megacities of China, and ship tracks along the coast and sources along the big rivers.

### 3.2 SO$_2$ trends over China

To construct time series of SO$_2$ we have averaged the data to annual means of the vertical columns derived from OMI. From these annual mean SO$_2$ data we constructed time series for each province (see Table A.1). Figure 4 shows the mean normalized time series for the 10 provinces with the highest total SO$_2$ column densities (i.e. Tianjin, Shandong, Hebei, Shanxi, Henan, Beijing, Jiangsu, Shanghai, Anhui and Liaoning), together responsible for 60% of all ambient SO$_2$ in China. The individual time series are drawn as thin black lines. The minimum and maximum of these time series for each year are shown in the grey shaded area to indicate the variability. The time series of Shanghai is the lowest black line of the 10 series, thus the reductions have been strongest in this province since 2005. Apart from Ningxia province, all provincial time series show very similar patterns. In general, the SO$_2$ concentrations were at a maximum in the year 2007, when the start of decreasing trend is visible in China. Despite some fluctuations the SO$_2$ concentrations remain relatively constant from 2010 till 2013, where after they are decreasing again.

A different trend is observed for Ningxia, a province in the mid-north of the country with a relative low population density and large coal resources. Here an increasing trend emerges for the years starting from 2010 when several new coal power plants were put into operation. A list of largest power plants (with a capacity of more than 600 MW) and the start year of their operation is shown in Table 3. From 2012 onward, the more stringent SO$_2$ emission regulations also started to have effect in Ningxia.

### 3.3 NO$_x$ emission trends over China

National NO$_x$ emission trends show a different pattern than those of SO$_2$. We observe an increasing trend till about 2012, with an exception of the year 2009 which is related to regulations started at the Olympic Games in 2008 (Mijling et al., 2008) and the global economic crisis which shortly slowed down the Chinese economic growth. Total NO$_x$ emissions in East China reached their peak levels in 2012, and have stopped increasing since this year. While the economy kept growing after 2012, the emission of NO$_x$ slowly decreases again as a result of the air quality regulations described in Section 1. According to the DECSO emission inversion, in 2015 the NO$_x$ emissions were 4.9 Tg N/yr, which is 22.8% lower than in the peak year 2012. However, the 2015 emissions were still 14.1% higher than in the reference year 2007. The trends per province (see Table A.2) show very similar patterns with only the starting year (the year with maximum NO$_x$ emissions) of the decrease in emissions varying over the provinces. Events like the Olympic Games in Beijing in 2008 and the World Expo in Shanghai in 2010, when temporary strict air quality regulations have been enforced, can be recognised in this Table as years with significant lower emissions for these provinces. In Figure 5a, the normalized (to the year 2007) time series of annual NO$_x$ emissions for East China (102-132°E, 18-50°N) is shown in similar way as for SO$_2$ in Figure 4. The mean, minimum and maximum of the 10 provinces with highest NO$_x$ emission are shown (Shandong, Hebei, Henan, Jiangsu, Guangdong, Shanxi, Zhejiang, Anhui, Sichuan and Hubei), together responsible for 65% of all Chinese NO$_x$ emissions. The thin black lines show the times series for the individual 10 provinces, where the lower line represents Guangdong. Figure 5b shows the peak year for each province. Provinces where air pollution regulations, for e.g. traffic, got a lot of attention at an early stage, like Beijing and Shanghai, have reached their maximum before 2011. Most industrialised regions show their peak in the years 2011-2013. Some of the less developed and populated provinces show a maximum in 2014, which means that their decrease in NO$_x$ emissions is very recent. Regional variations are mainly due to the fact that regulations for the NO$_x$ emission reductions, for instance in traffic or power plants, are determined and implemented on a provincial level (Liu et al., 2016b). For the province Ningxia we see a very similar pattern occurring as for SO$_2$, which shows that for this low-densely populated province traffic plays a small role and the trend is determined by the operation of newly-built power plants.

### 3.4 Air pollution in relation to fossil fuel consumption

To relate the observed SO$_2$ and NO$_x$ reduction to environmental regulations we have to take into account the coal and oil consumption in the same time period. The total coal consumption in Standard Coal Equivalent (SCE) units per year for China and the total oil consumption (also in SCE units) are shown in Figure 6, based on data of NBSC (2015). According to Guan et al. (2012) and Hong et al. (2016) the sum of the coal consumption of all provinces is more accurate than the number provided for the whole of China, thus we use the provincial totals for coal consumption. For NO$_x$ emissions the transport sector plays an important role, especially ships are one of the largest NO$_x$ emitters per fuel unit in the transport sector. The total freight transport almost doubles every 6 years in China.

Since the burning of coal and oil are the dominant sources of SO$_2$ and NO$_x$ emissions, we can consider the total emissions of these air pollutants as the product of the national use of coal and oil (activity) and the average emission factor of one unit coal/oil. The effectiveness of environmental regulation will be reflected in a decrease of this emission factor. Therefore, we divide the annual SO$_2$ column measured from satellites and the annual NO$_x$ emissions by the annual coal and oil consumption in China. In this way we get a measure of the emitted SO$_2$ or NO$_x$ per unit (SCE) of fossil fuel consumption reflecting the Chinese environmental policy. The results are shown in Figure 7. One might argue that SO$_2$ is more related to coal than oil, but division by only coal yields the same results. In our analysis we omit gas consumption since this is very limited in China and hence does not affect the results significantly.

We focus here mainly on the results for OMI, because of the instrument's high spatial resolution and lack of instrumental degradation. However, SO$_2$ data of the SCIAMACHY and GOME-2 instrument are also added in Figure 7 to be able to further look into the past (starting in 2003) and to verify the results of OMI. The SO$_2$ data of SCIAMACHY and GOME-2

are averaged over the summer months (April-September). The remaining monthly means are excluded from the analysis due to a lower accuracy at higher latitudes and a large part of the higher latitudes is missing due to snow cover. For OMI each data point is averaged over 12 months and the total area of China, which reduces the root-mean-square error to a negligible level. Biases among all instruments are removed by normalizing the values to those in reference year 2007. Up to 2009, the results agree fairly well. After 2009, we see the results of GOME-2 and OMI for $SO_2$ slowly diverge in time, which might be result of the instrument degradation of the UV spectra of GOME-2 after 2009 (Munro et al., 2016).

Changing weather conditions from year-to-year can affect the results for $SO_2$ concentrations and when these weather conditions are different during the overpass of SCIAMACHY and GOME-2 (around 9:30 local time) and overpass of OMI (around 13:30 local time), this can lead to differences between the instruments. The global coverage of SCIAMACHY is once every 6 days and for GOME-2 and OMI almost daily. The limited number of samples for SCIAMACHY makes this data more sensitive to weather conditions. Note that due to the nature of the inversion algorithm the $NO_x$ emission data is in general not sensitive to meteorological variability.

For $SO_2$ we see a big decrease in the years 2008 and 2009, while the desulphurization program of the 11[th] five-year plan started already in 2005/2006, when the authorities begin to reduce $SO_2$ emissions by installing desulphurization devices in many power plants (Lu et al., 2010). In 2006 $SO_2$ monitoring devices were also installed in the chimneys of the power plants. This resulted in a decrease in $SO_2$ emissions from 2006, while the much bigger decrease of $SO_2$ in 2008-2009 reflects the stronger government control at that time on the actual use of the equipment (Xu et al., 2011). After 2009, the $SO_2$ content per consumed coal unit only slowly decrease until 2011. From 2012 onwards we see a stronger annual decrease in $SO_2$. This coincides with the 12[th] five-year program when new measures were taken to upgrade the coal quality, to modernize the industry and to put more effort on law enforcement. Especially the law enforcement in the last years concerning the prohibition of flue gas bypass and the use of desulphurization devices in the steel industry played an important role.

For the $NO_x$ emissions the total annual emissions are used and divided in the same way as for $SO_2$ by the total coal and oil consumption. Here, however we should keep in mind that the transport sector (especially by shipping) emits much more $NO_x$ per fuel unit than the power and industrial sectors (see e.g. Zhao et al., 2013). Thus the percentage of the total fuel used by transport is relevant for the graph of $NO_x$. In the early years we see in general a small increase in $NO_x$ emissions per fuel unit due to the increasing fraction of the transport sector in the fuel use. Exceptions are the year 2009 and the recent year 2015. The year 2009 coincides with the global economic crisis (Lin and McElroy, 2011) when there was less export of goods from China. This affected especially the transport sector, mostly transport over water, as shown in De Ruyter de Wildt et al. (2012). Faber et al. (2012) and Boersma et al. (2015) showed that the economic crisis also resulted in a significant reduction of the average vessel speed to save fuel used by ship transport. This caused not only a shift in source sectors but in general led to lower $NO_x$ per fuel values. This explains the dip in pollution per fuel unit in 2009. After 2009 the $NO_x$ per fossil fuel is slowly increasing because the transport sector is growing faster than the energy sector and has a higher emission factor. Statistics of the NBSC (2015) show that the transport is growing with a factor two every 5-6 years (Wu et al., 2017). After 2012 the gradual increase of $NO_x$ per fuel slowly stops, and the year 2015 shows a sharp decline in $NO_x$ per fossil fuels unit. This can be directly related to the rapidly growing installation of SCR equipment at power plants since 2012 and to a lesser account to the introduction of new emission standards for cars, as shown by Liu et al. (2016). This strong reduction in $NO_x$ for 2015 and the equally strong reduction for $SO_2$ in 2014 and 2015 are a result of very effective recent environmental regulations in the last years in China. By comparing the efficiency level in 2015 with earlier levels we can conclude from Figure 7 that without these air quality regulations $SO_2$ concentrations would nowadays be about 2.5 times higher. For $NO_x$ per fossil fuel we were expecting a gradual growth after 2012 because of the continuing relative growth of the transport sector. Keeping this in mind we compare the years 2015 with 2012 and conclude that without air quality regulations the $NO_2$ concentrations would be at least 25% higher in China today.

On a provincial scale we can in principle do similar analyses, but unfortunately the provincial energy consumption related to coal and oil have a very high uncertainty due to inconsistencies in interprovincial imports and exports (Hong et al., 2016). We see this reflected in a high variability in the annual provincial data and sometimes missing data. Especially for the oil consumption the data have high uncertainties (Guan et al., 2011, Hong et al., 2016). Therefore, we have only analysed the 5 provinces with dominating coal consumption as shown in Figure 8. In this graph we excluded Guizhou province because of its difficult to interpret coal consumption in 2011 as a result of large power shortages (NBSC, 2015, Xinhua, 2011). For $SO_2$ per fossil fuel unit we see that all provinces follow the national trend. For $NO_x$ per fossil fuel we see more variation per province depending on the role of transport. Most of these by coal consumption dominated provinces start their decreasing trend from 2011 reflecting the national program on SCR installations starting that same year. It is interesting to see that the commissioning of new power plants in 2011 causes a strong increase in both $SO_2$ and $NO_x$ in the province Ningxia (see Figure 4 and 5a). However, when compensating for fossil fuel usage, one can see that the same national air quality regulations are applied here, as the trend in Figure 8 shows the same pattern as for other provinces. This show the strength of the presented method to assess the efficiency of air pollution regulations.

# 4   Discussion

The current developments in data products derived from satellite observations provide high quality time series of the air pollutants $NO_x$ and $SO_2$. Although the mean of observed $SO_2$ columns are not linearly related to the $SO_2$ emissions because of the influence of the weather, it can still be argued that these satellite data products, whether concentrations or emissions, provide a fair comparison over the various regions from year to year. By comparing these time series with fossil fuel energy consumption the economic growth is removed from the equation and we can monitor the effectiveness of air quality policies. We foresee that this method will become a valuable tool for policy makers concerning air quality regulations.

For China we see patterns in the trends of $SO_2$ that are similar for all provinces. In 2006 a nation-wide implementation of desulphurisation installations started. However, the effects are only visible in 2008 and 2009 when a strict control by the Chinese authorities on the actual use of the desulphurisation installations started. In 2009, we see the effect of the air quality regulations for $SO_2$ and $NO_x$ resulting from the global economic recession at the end of 2008. The increasing relative contribution of the transport sector to the $NO_x$ emission slowly increases the amount of $NO_x$ per fossil fuel unit after 2009. After 2011 we see a steadily decreasing $SO_2$ pollution per fossil fuel unit caused by various Chinese environmental regulations. In the last year of our time series, 2015, a clear effect becomes visible of very recent regulations for $NO_x$ emissions from power plants and heavy industry. The fit of linear trends often used in earlier studies is therefore no longer applicable to the Chinese situation.

For the first time it is shown from satellite observations that not only $NO_2$ concentrations are recently decreasing in China but also $NO_x$ emissions in all Chinese provinces are decreasing in the last two years. Showing this decreasing trend for emissions of $NO_x$ rules out any meteorological influences that affect concentrations. By the novel method of dividing these $NO_x$ emissions by the fossil fuel consumption and thereby showing the decreasing trend in emission factors in China, we also exclude the effect of economic changes, which has always been the driving factor in the trend of emissions in China in the past decades. We gave a complementary new overview of the main national air quality regulations in China from which changes in the emission factors of $NO_x$ and $SO_2$ can be understood and clarified. These trends in emission factors of $NO_x$ and $SO_2$ based on satellite observations might also be applied to verification of existing emission factors.

The availability of high quality satellite data for the last ten years is especially interesting for China where the situation is rapidly changing. For instance in Europe and Japan desulphurisation started much earlier when these satellite data were not yet available. On the other hand, in India $SO_2$ and $NO_x$ emissions are still growing and possible new regulations can be

monitored in the years to come with an even better quality using forthcoming sensors as e.g. TROPOMI on-board Sentinel-5 Precursor.

Despite the growing use of coal and oil in the last ten years in China we see reduced emissions per fuel unit in the past few years. This decreasing trend in both $SO_2$ and $NO_x$ for China is likely to continue in the coming years for which the Chinese national government has announced less use of coal, more environmental regulations for $SO_2$ and $NO_x$ and stricter reinforcement of control of environmental policies.

**Acknowledgements**

This research has been funded by the MarcoPolo project of the European Union Seventh Framework Programme (FP7/2007-2013) under Grant Agreement n° 606953 and by the GlobEmission project (Contract No 4000104001/11/I-NB) of the Data User Element programme of the European Space Agency.

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

**Figure 1: Location of power plants in China according to REAS v.2 (Kurokawa et al., 2013). The size of each dot indicates the emission of the power plants (Power plants in close proximity are combined in a single dot). In addition, a list is given of the provinces mentioned in this study.**

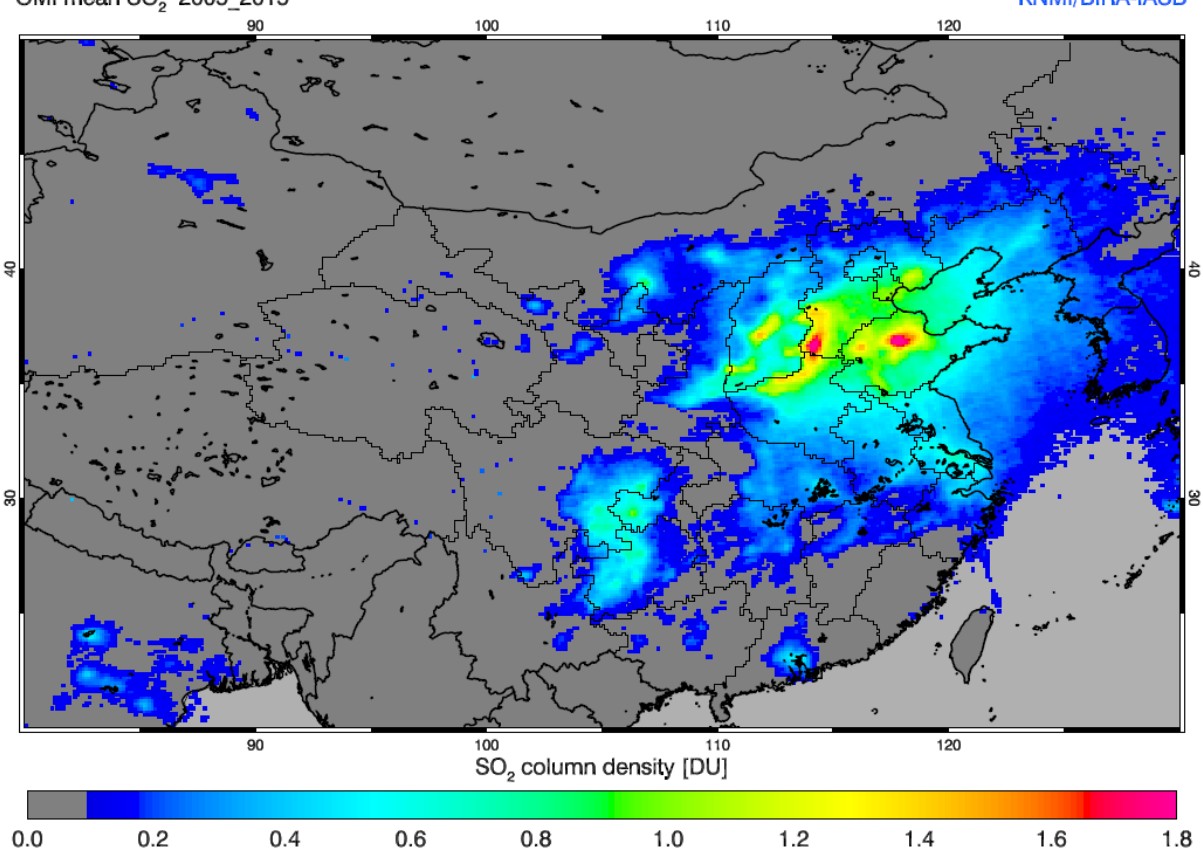

**Figure 2: Average SO$_2$ concentrations for the period 2005 to 2015 as observed by the OMI satellite instrument. Data below 0.1 DU is masked (grey colour).**

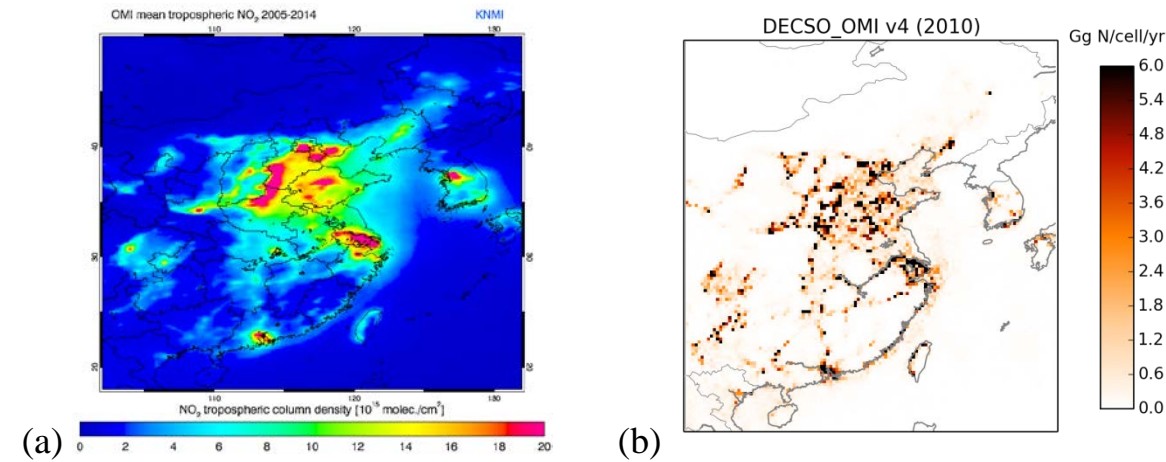

5          (a)                                              (b)

**Figure 3: (a) The averaged tropospheric NO$_2$ concentrations over China measured by OMI in the period 2005-2014. (b) The NO$_x$ emissions in the year 2010 derived from the OMI satellite observations.**

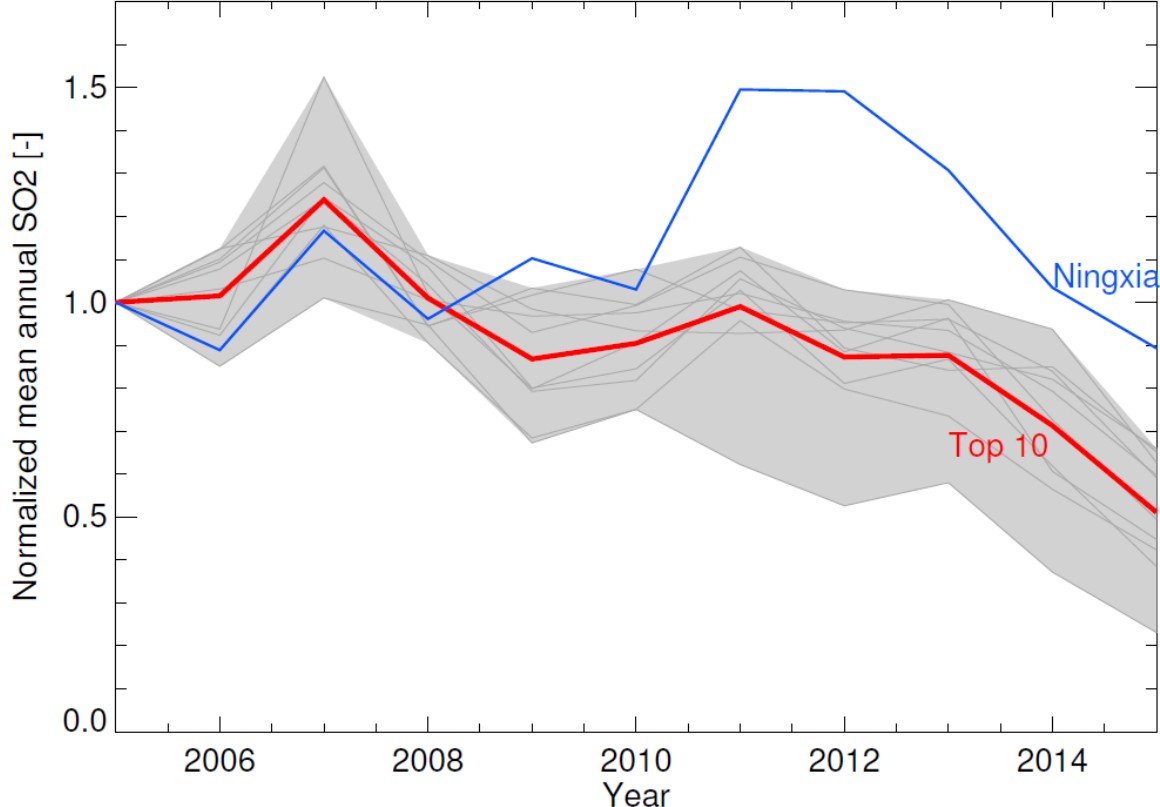

**Figure 4: Time series (red line) of the annual mean of the 10 provinces with the highest SO$_2$ concentrations derived from the OMI satellite observations. The time series are normalized to their value in 2005. The grey area indicates the maximum range of the individual values of the times series of each of the 10 provinces. The thin black lines show the individual time series of those provinces. The province of Ningxia has a distinct deviating trend, here shown in blue.**

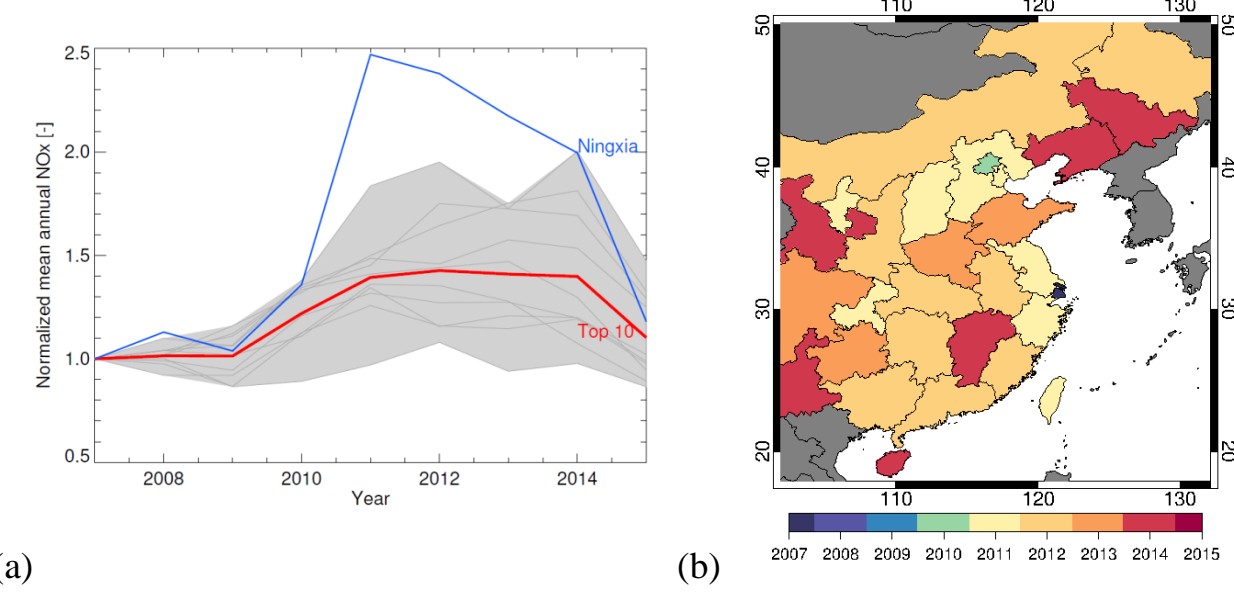

(a)                                                       (b)

**Figure 5: (a) Shown are the annual total NO$_x$ emission estimates for the last 9 years for the top 10 of highest NO$_x$ emitting provinces in East China. Emissions are derived with DECSO V4 using OMI observations. The thin black lines show the individual time series of those provinces. (b) Peak year of the NO$_x$ emissions per province.**

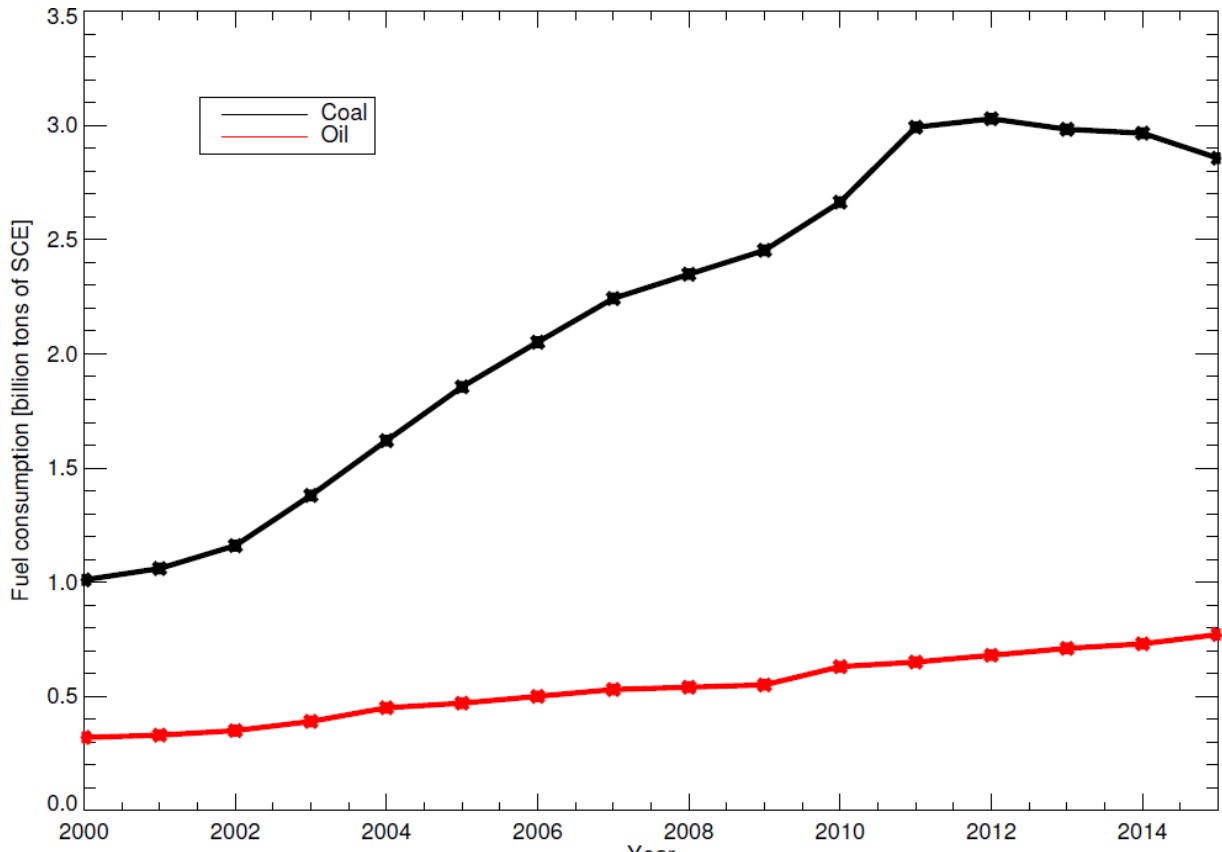

**Figure 6: In black the annual coal consumption and in red the annual oil consumption for China is shown.**

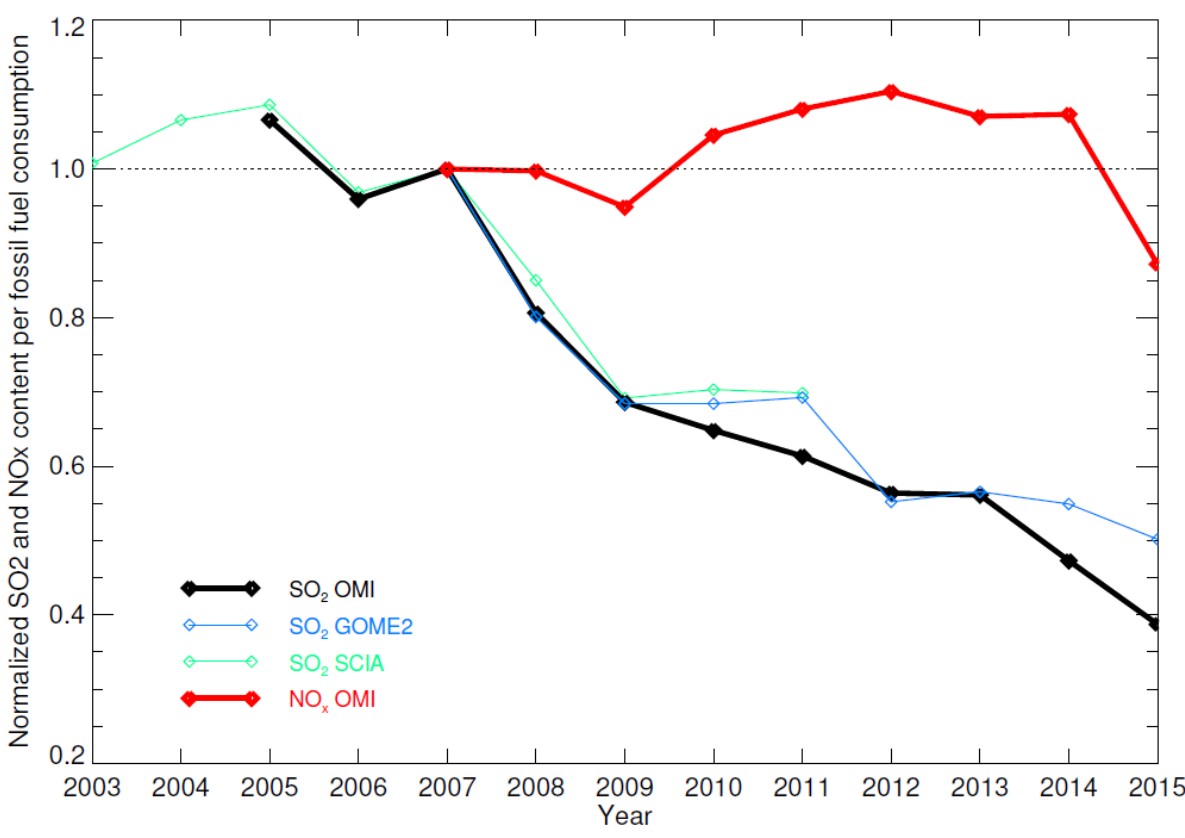

**Figure 7: Time series of the ratio of the mean SO$_2$ columns and the fossil fuel consumption in China based on observations of OMI (black), SCIAMACHY (green), and GOME-2 (blue). The ratios of the annual NO$_x$ emissions and the fossil fuel consumption is based on observations of OMI (red). All time series are normalized to the year 2007.**

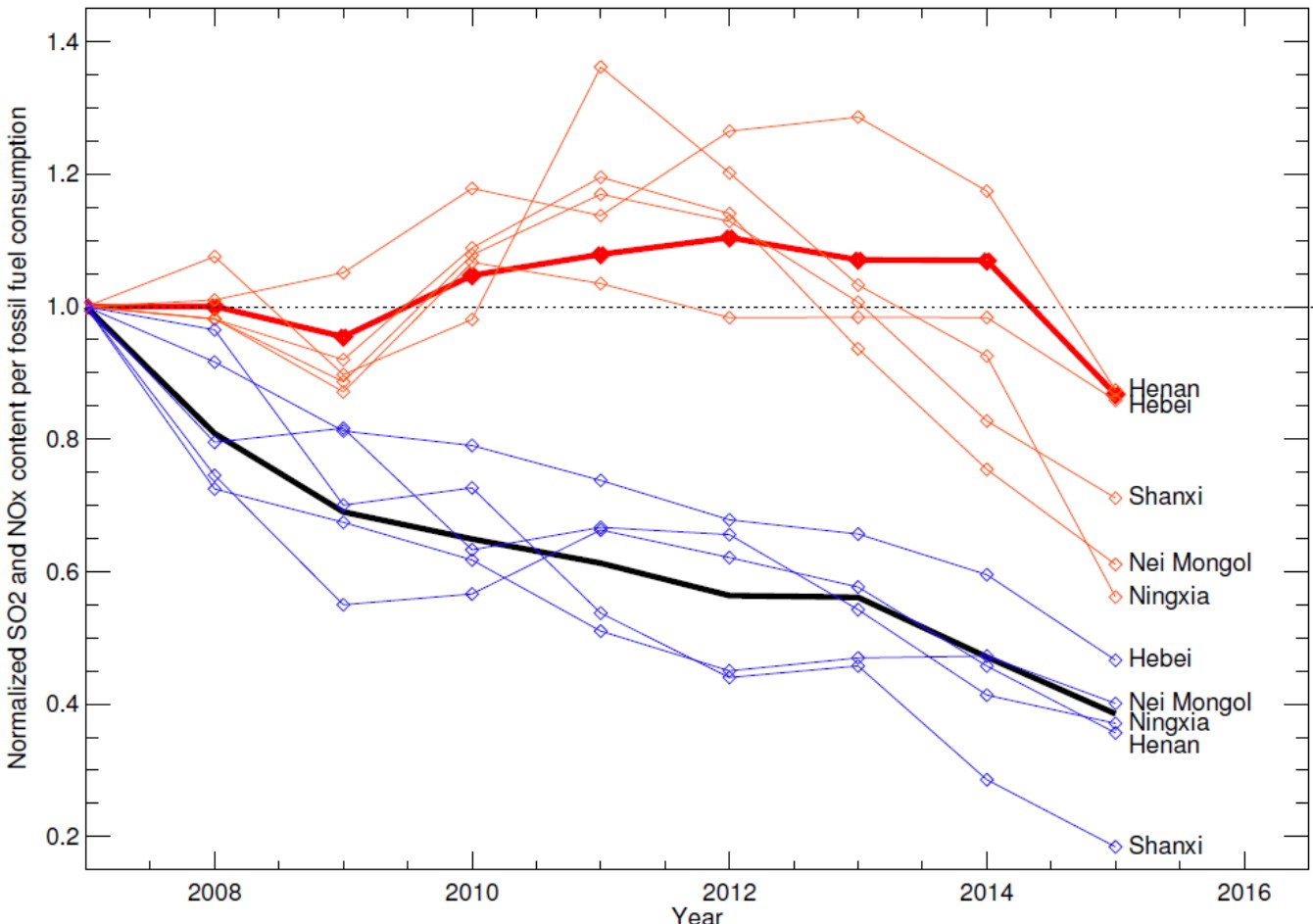

5 **Figure 8: Same graph as Figure7 but with time series for the provinces Hebei, Henan, Nei Mongol, Ningxia and Shanxi included. Time series per province of the ratio of the mean SO$_2$ columns and the fossil fuel consumption are drawn in blue. The ratios of the annual NO$_x$ emissions and the fossil fuel consumption per province are shown in red. All time series are normalized to the year 2007 and based on OMI observations.**

**Table 1: Environmental regulations of the Chinese national government to reduce SO$_2$ in the air.**

| Start year of implementation | Regulation | Reference |
|---|---|---|
| 2005-2006 | Desulphurization techniques in power plants. | Li et al., 2010 |
| 2005-2012 | Closure of several of the most polluting power plants | Liu et al., 2015 |
| 2008 | Stricter control of implementation of desulphurization in power plants | Xu et al., 2011 Liu et al., 2015 |
| 2011 | Use of more gas and renewable energies instead of coal | NBSC, 2015 |
| January 2012 | New emission standard of air pollutants for thermal power plants | MEP, 2015 |
| 2013 | Mandatory SO$_2$ filtering of small-scale coal-fired industry | Zhang, 2013, NDRC, 2013 |
| End of 2013 | Stricter control of environmental policy | CAAC, 2013, State Council, 2014 |
| End of 2013 | Further desulphurization in industry | CAAC, 2013, NDRC, 2013 |
| 2014 | Phasing out small-scale coal-fires boilers | CAAC, 2013, State Council, 2014 |
| 2014 | Closure of 2000 small-scale coal mines | Zhu, 2013 |
| End of 2014 | Use of low-sulphur coal | State Council, 2014 |
| End of 2014 | Cap on coal consumption | State Council, 2014 |

**Table 2: Environmental regulations of the Chinese national government to reduce NOx emissions.**

| Year of implementation | Regulation | Reference |
|---|---|---|
| 2011-2015 | Installation of Selective Catalytic Reduction (SCR) equipment at power plants. In 2013 the SCR equipment was installed in about 50% of all power plants. | Liu et al., 2016, CAAC, 2013 |
| 2007 | China 3 (Euro 3) emissions standards for cars, nationwide | Wu et al., 2017 |
| 2011 | China 4 (Euro 4) emissions standards for gasoline cars, nationwide. | Wu et al., 2017 |
| 2015 | China 4 (Euro 4) emissions standards for diesel cars, nationwide. | Wu et al., 2017 |

**Table 3: Main power plants in Ningxia province (> 600 MW). Data collected from www.sourcewatch.org.**

| Power plant | Capacity (MW) | In operation since | Remark |
|---|---|---|---|
| CPI Linhezhen | 700 | unknown | |
| Daba-1 | 1200 | < 2000 | |
| Daba-2 | 1100 | unknown | An extension of Daba-1 |
| Ningxia Zhongning-2 | 660 | 2005-2006 | |
| Guodian Shizuishan-2 | 1980 | 2006 | |
| Ningdong Maliantai | 660 | 2006 | |
| Huadian Ningxia Lingwu units 1&2 | 1200 | 2007 | |
| Guodian-Dawukou | 1100 | 2010 | Extension of the original 440 MW plant |
| Guohua Ningdong | 660 | 2010 | |
| Ningxia Liupanshan | 660 | 2010 | |
| Huadian Ningxia Lingwu units 3&4 | 2120 | 2010-2011 | |
| Shenhua Yuanyang Lake | 1320 | 2010-2011 | |
| Shuidonggou | 1200 | 2011 | |
| Ningdong Younglight | 660 | 2013 | |

**Appendix A**

**Table A1 Annual SO₂ column densities (DU/grid cell) per province observed by OMI**

| Province | 2005 | 2006 | 2007 | 2008 | 2009 | 2010 | 2011 | 2012 | 2013 | 2014 | 2015 |
|---|---|---|---|---|---|---|---|---|---|---|---|
| Anhui | 0.625 | 0.586 | 0.953 | 0.637 | 0.499 | 0.567 | 0.671 | 0.553 | 0.602 | 0.379 | 0.280 |
| Beijing | 0.753 | 0.829 | 0.989 | 0.711 | 0.778 | 0.749 | 0.850 | 0.673 | 0.634 | 0.640 | 0.491 |
| Chongqing | 0.514 | 0.509 | 0.530 | 0.567 | 0.580 | 0.580 | 0.492 | 0.370 | 0.469 | 0.269 | 0.136 |
| Fujian | 0.099 | 0.123 | 0.196 | 0.135 | 0.104 | 0.113 | 0.112 | 0.080 | 0.107 | 0.076 | 0.064 |
| Gansu | 0.144 | 0.136 | 0.150 | 0.130 | 0.135 | 0.123 | 0.134 | 0.127 | 0.131 | 0.105 | 0.103 |
| Guangdong | 0.251 | 0.257 | 0.280 | 0.239 | 0.171 | 0.177 | 0.138 | 0.095 | 0.118 | 0.086 | 0.080 |
| Guangxi | 0.199 | 0.203 | 0.270 | 0.236 | 0.127 | 0.190 | 0.179 | 0.092 | 0.134 | 0.091 | 0.072 |
| Guizhou | 0.424 | 0.478 | 0.532 | 0.516 | 0.418 | 0.424 | 0.357 | 0.261 | 0.345 | 0.167 | 0.100 |
| Hainan | 0.098 | 0.086 | 0.091 | 0.092 | 0.060 | 0.090 | 0.106 | 0.027 | 0.087 | 0.055 | 0.000 |
| Hebei | 0.903 | 0.931 | 0.996 | 0.908 | 0.874 | 0.881 | 0.922 | 0.863 | 0.844 | 0.716 | 0.540 |
| Heilongjiang | 0.134 | 0.141 | 0.144 | 0.142 | 0.124 | 0.138 | 0.154 | 0.162 | 0.125 | 0.134 | 0.135 |
| Henan | 1.036 | 0.920 | 1.222 | 0.938 | 0.709 | 0.778 | 0.992 | 0.827 | 0.762 | 0.585 | 0.439 |
| Hubei | 0.487 | 0.477 | 0.603 | 0.490 | 0.342 | 0.386 | 0.479 | 0.365 | 0.378 | 0.288 | 0.176 |
| Hunan | 0.364 | 0.330 | 0.448 | 0.371 | 0.270 | 0.281 | 0.320 | 0.240 | 0.259 | 0.180 | 0.112 |
| Jiangsu | 0.847 | 0.782 | 1.054 | 0.917 | 0.678 | 0.716 | 0.871 | 0.687 | 0.735 | 0.524 | 0.326 |
| Jiangxi | 0.272 | 0.278 | 0.373 | 0.267 | 0.202 | 0.222 | 0.244 | 0.197 | 0.230 | 0.184 | 0.136 |
| Jilin | 0.205 | 0.233 | 0.260 | 0.259 | 0.191 | 0.207 | 0.201 | 0.187 | 0.203 | 0.192 | 0.156 |
| Liaoning | 0.512 | 0.576 | 0.602 | 0.568 | 0.504 | 0.478 | 0.475 | 0.479 | 0.515 | 0.480 | 0.321 |
| NeiMongol | 0.154 | 0.170 | 0.200 | 0.174 | 0.175 | 0.180 | 0.191 | 0.178 | 0.177 | 0.186 | 0.152 |
| Ningxia | 0.234 | 0.208 | 0.273 | 0.225 | 0.258 | 0.241 | 0.350 | 0.349 | 0.306 | 0.242 | 0.209 |
| Qinghai | 0.079 | 0.085 | 0.077 | 0.080 | 0.079 | 0.088 | 0.083 | 0.091 | 0.096 | 0.082 | 0.086 |
| Shaanxi | 0.357 | 0.301 | 0.401 | 0.324 | 0.261 | 0.269 | 0.338 | 0.304 | 0.315 | 0.246 | 0.224 |
| Shandong | 1.197 | 1.309 | 1.531 | 1.315 | 1.113 | 1.188 | 1.323 | 1.232 | 1.191 | 0.870 | 0.592 |
| Shanghai | 0.874 | 0.744 | 0.883 | 0.828 | 0.588 | 0.656 | 0.544 | 0.460 | 0.507 | 0.325 | 0.202 |
| Shanxi | 0.748 | 0.806 | 0.928 | 0.779 | 0.593 | 0.612 | 0.789 | 0.703 | 0.661 | 0.614 | 0.493 |
| Sichuan | 0.429 | 0.429 | 0.513 | 0.376 | 0.415 | 0.427 | 0.394 | 0.293 | 0.350 | 0.198 | 0.123 |
| Taiwan | 0.089 | 0.071 | 0.081 | 0.085 | 0.074 | 0.090 | 0.074 | 0.055 | 0.086 | 0.052 | 0.051 |
| Tianjin | 1.197 | 1.344 | 1.577 | 1.132 | 1.217 | 1.289 | 1.176 | 1.140 | 1.150 | 1.005 | 0.708 |
| XinjiangU. | 0.073 | 0.074 | 0.087 | 0.093 | 0.088 | 0.094 | 0.090 | 0.090 | 0.111 | 0.101 | 0.084 |
| Xizang/Tibet | 0.080 | 0.097 | 0.086 | 0.091 | 0.094 | 0.111 | 0.096 | 0.097 | 0.083 | 0.087 | 0.087 |
| Yunnan | 0.140 | 0.159 | 0.182 | 0.147 | 0.144 | 0.153 | 0.149 | 0.131 | 0.127 | 0.100 | 0.085 |
| Zhejiang | 0.383 | 0.337 | 0.452 | 0.395 | 0.297 | 0.316 | 0.403 | 0.258 | 0.321 | 0.212 | 0.158 |
| P.R.China | 0.397 | 0.392 | 0.444 | 0.373 | 0.330 | 0.342 | 0.358 | 0.335 | 0.332 | 0.280 | 0.225 |

**Table A2 Annual NO$_x$ emissions (Gg N/year) per province in the domain of DECSO (in parenthesis the fraction of provincial area considered) derived from OMI observations**

| Province | 2007 | 2008 | 2009 | 2010 | 2011 | 2012 | 2013 | 2014 | 2015 |
|---|---|---|---|---|---|---|---|---|---|
| Anhui | 167 | 169 | 187 | 224 | 242 | 292 | 288 | 282 | 215 |
| Beijing | 91 | 62 | 90 | 107 | 88 | 80 | 89 | 74 | 64 |
| Chongqing | 54 | 57 | 70 | 75 | 87 | 95 | 96 | 100 | 70 |
| Fujian | 96 | 114 | 100 | 114 | 161 | 162 | 153 | 167 | 137 |
| Gansu (61%) | 31 | 38 | 37 | 42 | 61 | 73 | 60 | 78 | 52 |
| Guangdong | 383 | 383 | 331 | 341 | 371 | 413 | 360 | 374 | 331 |
| Guangxi | 118 | 148 | 118 | 145 | 152 | 224 | 224 | 200 | 157 |
| Guizhou | 107 | 130 | 142 | 154 | 131 | 194 | 191 | 180 | 122 |
| Hainan | 8 | 13 | 11 | 17 | 23 | 22 | 25 | 37 | 30 |
| Hebei | 427 | 423 | 403 | 515 | 563 | 543 | 544 | 511 | 436 |
| Heilongjiang(74%) | 33 | 36 | 30 | 25 | 43 | 54 | 40 | 49 | 31 |
| Henan | 334 | 347 | 370 | 445 | 470 | 481 | 491 | 433 | 315 |
| Hubei | 135 | 140 | 144 | 186 | 248 | 263 | 233 | 270 | 199 |
| Hunan | 109 | 112 | 124 | 162 | 163 | 216 | 184 | 218 | 187 |
| Jiangsu | 374 | 344 | 344 | 421 | 470 | 433 | 428 | 445 | 365 |
| Jiangxi | 51 | 58 | 65 | 73 | 85 | 105 | 111 | 150 | 112 |
| Jilin | 30 | 22 | 18 | 20 | 45 | 50 | 43 | 48 | 43 |
| Liaoning | 122 | 128 | 124 | 169 | 205 | 225 | 178 | 199 | 173 |
| NeiMongol (83%) | 98 | 116 | 117 | 156 | 215 | 215 | 169 | 142 | 111 |
| Ningxia | 30 | 34 | 32 | 41 | 75 | 72 | 66 | 61 | 36 |
| Shaanxi | 118 | 118 | 113 | 181 | 216 | 222 | 196 | 208 | 158 |
| Shandong | 464 | 510 | 493 | 629 | 689 | 677 | 731 | 712 | 580 |
| Shanghai | 96 | 103 | 101 | 95 | 109 | 75 | 83 | 93 | 84 |
| Shanxi | 292 | 284 | 253 | 328 | 397 | 395 | 373 | 313 | 260 |
| Sichuan (51%) | 155 | 158 | 179 | 204 | 232 | 254 | 271 | 280 | 205 |
| Taiwan | 100 | 106 | 98 | 106 | 113 | 118 | 106 | 114 | 111 |
| Tianjin | 77 | 86 | 99 | 136 | 152 | 114 | 102 | 97 | 88 |
| Yunnan (36%) | 83 | 109 | 105 | 97 | 118 | 159 | 144 | 176 | 126 |
| Zhejiang | 243 | 253 | 247 | 270 | 327 | 281 | 294 | 292 | 240 |
| EastChina | 4332 | 4502 | 4454 | 5382 | 6150 | 6402 | 6179 | 6201 | 4941 |