# Peer review of "Cleaning up the air: Effectiveness of air quality policy for $SO_2$ and $NO_x$ emissions in China"

_Atmospheric Chemistry and Physics, 2016_

## Referee Comment (RC1) · Anonymous Referee #1 · 4 Jun 2016

While I find the topic of the manuscript timely and intriguing, I think that the paper is too thin on content and there is little independent information presented or literature citations given to support their weak conclusions. For instance, the authors do not really discuss or quantify the "effectiveness of air quality policy", as given in the title, beyond saying that the trends in space-based air quality data appear somewhat consistent. Further analysis is needed before this manuscript should be published or before it can live up to what is promised by the title.

Comments

Abstract. Please clearly say in the abstract what is new and interesting about your work. "unprecedented accuracy" – you didn't show this or even really discuss this. The

abstract does not have any interesting or strong conclusions.

Introduction. 1st paragraph. 1st sentence. Why are satellite instruments "especially effective"? 2nd sentence. What improved datasets? 4th sentence. How is this data set "improved"? Accuracy?

Please put your work into the context of other studies of emissions and trends over China. There are quite a few recent ones to discuss that use satellite data. Please include the new paper by McLinden et al. (Nature). Many of these discuss individual emission sources, such as power plants. However, you do not, which would be necessary to estimate the effectiveness of environmental regulations.

Section 3.2. There are many speculations here. Back up them up with provincial data. Please overplot fuel consumption data in Figure 3.

Section 3.3. Are the trends in OMI NO2 consistent with the provincial emissions data? Please plot.

Section 3.4. Again, I'm interested in the provincial data.

Figures 1 & 2. Need to have a map of provinces for the reader to refer to with major cities. Most readers will not know provincial names. It may also help to plot the locations of major power plants.

---

## Author Comment (AC1) · 18 Jun 2016

**Response to referee report #1**

We thank Referee #1 for the valuable comments. Below we give a point-to-point response to comments by referee. The referee's comments are given in italic font, while our response is given in bold font.

*While I find the topic of the manuscript timely and intriguing, I think that the paper is too thin on content and there is little independent information presented or literature citations given to support their weak conclusions.*

**In our paper we conclude that NOx emissions would have been 30 % higher and SO2 concentrations would be 3 times higher as they are today without the strong air quality regulations taken in the last 10 years. This is not a weak conclusion for a country with such serious air pollution problems. The conclusions come from careful analysis of satellite data as shown in our paper. All independent sources are properly referenced.**

*For instance, the authors do not really discuss or quantify the "effectiveness of air quality policy", as given in the title, beyond saying that the trends in space-based air quality data appear somewhat consistent.*

**The effectiveness is literally quantified as 30 % and factor 3 for respectively NOx and SO2. As shown in the paper this reduction is only an effect of air quality regulations and excluding all economic effects. It is not clear to us what text the referee refers to with "***trends in space-based air quality data appear somewhat consistent.***".**

*Further analysis is needed before this manuscript should be published or before it can live up to what is promised by the title.*

**Please, specify what analysis you feel is missing.**

*Comments*

*Abstract. Please clearly say in the abstract what is new and interesting about your work.*

**New in this paper is that the effect of air quality measures is quantified by dividing the air pollutant emissions by the fossil fuel use, which leads to a trend in emission factor measured from space. We will emphasize this in the update of the paper. This can be done successfully because satellite products and algorithms have improved greatly. Satellite measurement indicate independently that NOx emissions in China peaked in 2012 and are greatly reduced since.**

*"unprecedented accuracy" – you didn't show this or even really discuss this.*

**The term "unprecented" indeed sounds somewhat dramatic, but it only states that our new data set improves over old ones. We will replace this with the more neutral "improved". The improved accuracy is mentioned and further discussed in Theys et al. (2015) and Ding et al. (2015, 2016).**

*The abstract does not have any interesting or strong conclusions.*

**See comments above.**

*Introduction. 1st paragraph. 1st sentence. Why are satellite instruments "especially effective"?*

**We agree the wording is not entirely logic and we will change the sentence into:**

**" Satellite instruments can monitor air quality from space by mapping e.g. aerosols and tropospheric ozone, but are especially useful for emission estimates in observing the relatively short-living gases nitrogen dioxide (NO2) and sulphur dioxide (SO2)"**

*2nd sentence. What improved datasets? 4th sentence. How is this data set "improved"? Accuracy?*

**The Improved datasets are directly discussed after this sentence in the same paragraph. SO2 from OMI described in Theys et al. (2015), and NOx emissions from OMI with DECSO v4, described in Ding et al. (2015, 2016). The accuracy is described in these papers. For SO2 the column concentrations are on average within 12 % in agreement with ground observations. NOx emissions have an accuracy of about 20% in each grid cell. We will add this information to the text.**

*Please put your work into the context of other studies of emissions and trends over China. There are quite a few recent ones to discuss that use satellite data. Please include the new paper by McLinden et al. (Nature). Many of these discuss individual emission sources, such as power plants. However, you do not, which would be necessary to estimate the effectiveness of environmental regulations.*

**McLinden et al. (Nature) was published 1 week after we submitted this paper, but we will of course refer and discuss this work in the update of our paper. Other studies about emissions and trends over China that are referred or discussed in our paper are Richter et al. (2005), van der A et al. (2006), Stavrakou et al. (2008), Kurokawa et al. (2009), Lee et al. (2010), Li et al (2011), He H. et al. (2012), Zhang et al. (2012), Yang et al. (2013), Mijling et al. (2013), Fioletov et al. (2015), Krotkov et al. (2015), and Liu et al. (2016).**

**Indeed we discuss the emissions on a provincial level to give an overview for a country which harbours at least 1000 power plants. A single power plant is therefore not representative for the air quality regulations. That is why we use the total fossil fuel use in combination with satellite observations. We present a new method to estimate the total effectiveness of environmental**

**regulations, by directly relating the emissions to the fossil fuel consumption in the country. This top-down approach saves us the effort of evaluating all individual contributions.**

*Section 3.2. There are many speculations here. Back up them up with provincial data. Please overplot fuel consumption data in Figure 3.*

**Section 3.2  Unfortunately the fuel consumption data for Chinese provinces is not publicly available, which limits our possibilities. However, many regulations are nation-wide as are our main analysis and conclusions. The fuel consumption data is shown in Figure 5.**

*Section 3.3. Are the trends in OMI NO2 consistent with the provincial emissions data? Please plot.*

**Section 3.3 We have done our analysis on the NOx emission data from inversions and not on NO2 concentrations, since emissions are clearly localised while concentrations are strongly affected by transport and meteorology. The quality of the inversion has been assessed in Mijling et al. [2013]. Also emissions are directly affected by air quality regulations.**

*Section 3.4. Again, I'm interested in the provincial data.*

**Unfortunately the fuel consumption data for provinces is not publicly available. Perhaps the reviewer is aware of sources we overlooked?**

*Figures 1 & 2.  Need to have a map of provinces for the reader to refer to with major cities.  Most readers will not know provincial names.  It may also help to plot the locations of major power plants.*

**We will add a map with the provinces of China (and power plant density) for convenience of the reader.**

---

## Referee Comment (RC2) · Anonymous Referee #2 · 3 Aug 2016

The authors present a study of temporal trends of NO2 and SO2 derived from satellite observations over China, and relate them directly to changes in fossil fuel consumption in order to investigate the effectiveness of environmental regulations. While the former has been done in several studies before, the latter is to my knowegedge new and provides an interesting approach. The study thus matches the scope of ACP and should generally be published.

However, the provided material is rather sparse (in particular as the introduction let the reader expect to see an analysis on provincal level, which is not given), and the results for NOx are not convincing; the authors try to interpret some local maxima by some reason, but I see no consistent explanation for the whole, rather complex, temporal

pattern.

Thus, the study needs major revisions, in particular for NOx, providing additional information which either substantiates the discussion of trends or let the authors be more cautious with their statement about NOx concentrations being 30% higher without regulations.

Major concerns:

- Provincial levels The authors point out that it is great to have improved NO2 and SO2 datasets on high spatial resolution, which allow the analysis of time series on provincial level. Thus, the authors should indeed investigate the trends of SO2, NOx, fossil fuel, and ratios on provinical levels, which probably will provide valuable further information and help to understand/assess the NOx-per-fuel trend (see below).

- Annual vs. semi-annual means: For SO2, only April-September is considered "due to a lower accuracy at higher latitudes" (3.2) For NO2, information on the kind of averaging is missing in 3.3, but later it is stated that total annual emissions are used (3.4). Why? The lower accuracy and snow/ice argument holds as well for NO2. As the trends for both SO2 and NO2 are compared to fossil fuel and to each other, the period for calculating means has to be consistent. In any case, the authors should also provide the "winter"-trend for SO2, despite the lower accuracy. Is it similar (with higher noise) or significantly different from Fig. 3?

- NOx regulations Almost no information is provided about the concrete NOx regulations. Please discuss the different possibilities in general, and the taken measures in detail, for reducing NOx over China, and provide a table similar as for SO2. From what I learned from the media, there were different measures taken during the Olympic games, like shutting off power plants nearby and building new ones more remotely (which would change the local, but not the total trends, underlining the need for investigations on provincal levels), or reduction of traffic (which would affect the NOx, but not the NOx per fuel). These (and other) different measures and their effect on NOx vs.

NOx per fuel trends have to be discussed.

- NOx per fossil fuel While the SO2 per fuel significantly decreases over the years, in accordance to regulations, the situation is less clear for NOx. The authors state that already 2008 the regulations worked out (explaining the 2009 minimum). But why is NOx per fuel increasing again (by 20%!) in the following years? Have the measures been cancelled? The attempt to explain Fig. 6 by shipping is pure speculation, as it is not supported by any data. The conclusion that NO2 would be 30% higher today without the measures taken is not convincing unless the decline in 2015 compared to the high plateau 2011-2014 is explained; or were all measures concerning NOx just taken in 2015? In any case, the 30% is overestimated as it compares the minimum to the maximum of the timeseries, completely ignoring statistical fluctuations.

Minor comments:

Page 1 Line 13: What does "spatially consistent" mean?

Page 1 Line 25: The factor of 3 for SO2 is different from the statement in 3.4, and not supported by the presented data.

Page 1 Line 31: "concentrations" should be "column densities"

Page 2 Line 11: Please provide the full name of "He K."

Page 3 Line 2: Add a reference to Liu et al., 2016: http://www.atmos-chem-phys.net/16/5283/2016/

Page 3 Line 12: Please provide some information how and how far the new retrieval improves the quality of SO2.

Page 5 Line 15: NO2 changes during the Olympic games (or during the Shanghai Expo) have been discussed before; please add references.

Page 6 Line 25: "by definition not sensitive": This is a too strong statement which only holds under the assumption that CHIMERE is doing everything right within the DESCO

algorithm.

Page 7 Line 6: There is no "Fig. 5b".

---

## Author Comment (AC2) · 15 Sep 2016

**Response to referee report #2**

We thank Referee #2 for the valuable comments. Below we give a point-to-point response to comments by referee. The referee's comments are given in italic font, while our response is given in bold font.

*The authors present a study of temporal trends of NO2 and SO2 derived from satellite observations over China, and relate them directly to changes in fossil fuel consumption in order to investigate the effectiveness of environmental regulations. While the former has been done in several studies before, the latter is to my knowledge new and provides an interesting approach. The study thus matches the scope of ACP and should generally be published.*

*However, the provided material is rather sparse (in particular as the introduction let the reader expect to see an analysis on provincal level, which is not given), and the results for NOx are not convincing; the authors try to interpret some local maxima by some reason, but I see no consistent explanation for the whole, rather complex, temporal pattern.*

*Thus, the study needs major revisions, in particular for NOx, providing additional information which either substantiates the discussion of trends or let the authors be more cautious with their statement about NOx concentrations being 30% higher without regulations.*

**The temporal patterns in NOx emissions can be explained by the (lack of) regulations on NOx emissions, a shift in fuel consumption patterns and other economic factors. We hope to have explained the trends and our conclusions better in the revised version of our paper by adding more discussion on the results. To support our conclusions and analysis we have added in the revised version the provincial data for SO2 concentrations and NOx emissions per year and province. In order to clarify our findings we have revised the Figures of the time series by adding the provincial time series of the 10 most emitting provinces. More details about our revisions follow below.**

**Major concerns:**

*- Provincial levels The authors point out that it is great to have improved NO2 and SO2 datasets on high spatial resolution, which allow the analysis of time series on provincial level. Thus, the authors should indeed investigate the trends of SO2, NOx, fossil fuel, and ratios on provincial levels, which probably will provide valuable further information and help to understand/assess the NOx-per-fuel trend (see below).*

**The NOx and SO2 datasets are indeed analysed on a provincial level. To summarize our findings we decided not to show the individual time series for 30 provinces, but instead show an average and range of the timeline for the 10 most polluted provinces. Individual time series have now been added in tables for SO2 and NOx with the concentration and emissions per year and per provinces. In the text we elaborated more on the provincial time series.**

The fuel consumption per province was not publicly available at the time of submitting our paper. Only recently the provincial data became available for most provinces, although we have no indication of the uncertainty of these numbers, especially for the smallest provinces. The ratio per province will combine the uncertainties on these coal/oil consumption numbers with the errors in the satellite data. This makes it difficult to draw any conclusions out of these ratios and therefore we have decided to ignore the provincial fuel numbers in this study.

On the other hand, we found evidence in the papers of Guan et al. (2012) and Hong et al. (2016) that the sum of the coal consumption of all provinces is more accurate than the number provided for the whole of China, that is why we have updated Figure 5 and 6 with the total coal consumption of the provinces, which only slightly changes the results. For oil we have not enough provincial data to do the same.

Guan, D., Z. Liu, Y. Geng, S. Lindner and K. Hubacek, The gigatonne gap in China's carbon dioxide inventories, Nature Climate Change, 2, 672–675, (2012), doi:10.1038/nclimate1560

Hong, C., Zhang, Q., He, K., Guan, D., Li, M., Liu, F., and Zheng, B.: Variations of China's emission estimates response to uncertainties in energy statistics, Atmos. Chem. Phys. Discuss., doi:10.5194/acp-2016-459, in review, 2016.

*- Annual vs. semi-annual means: For SO2, only April-September is considered "due to a lower accuracy at higher latitudes" (3.2) For NO2, information on the kind of averaging is missing in 3.3, but later it is stated that total annual emissions are used (3.4). Why? The lower accuracy and snow/ice argument holds as well for NO2. As the trends for both SO2 and NO2 are compared to fossil fuel and to each other, the period for calculating means has to be consistent. In any case, the authors should also provide the "winter"-trend for SO2, despite the lower accuracy. Is it similar (with higher noise) or significantly different from Fig. 3?*

It is true that both NO2 and SO2 are irregularly and sparsely sampled by the satellite at high latitudes in wintertime. This lowers the accuracy and potentially introduces biases when averages are calculated for these periods based on the available satellite measurements. The SO2 analysis in this paper depends on such averages. NOx emissions, however, are derived using an inversion algorithm insensitive to data gaps.

. The SO2 data from OMI is worse in winter than in summer, but the annual data is good enough and not significantly different from Figure 3. The reason we switched to summer means is that the SCIAMACHY SO2 observations are very scarce in winter, since in the most optimal situation SCIAMACHY has global coverage only once every 6 days. On top of that the SCIAMACHY data is more noisy due to the lower spatial resolution. To keep consistency for all SO2 data we had done all analysis for summer means. However, triggered by and based on the remarks of the reviewer about consistency, the authors came to the conclusion that it is better to use annual averages for both OMI products, since this is the main focus of our study and these data are compared to annual coal/oil consumptions.

For SCIAMACHY and GOME-2 we still use summer values. If we switch to annual means the GOME-2 line does not change a lot, but for SCIAMACHY the changes are much larger, although the

decreasing trend clearly remains. For SCIAMACHY there is practically no data above 45 degree North and in other regions the number of data point is still scarce due to the global coverage once every 6 days. This makes the SCIAMACHY data very sensitive to weather effects.

We have adapted the text accordingly in section 3.2 and 3.3 and made new Figures 3 and 6.

*- NOx regulations Almost no information is provided about the concrete NOx regulations. Please discuss the different possibilities in general, and the taken measures in detail, for reducing NOx over China, and provide a table similar as for SO2. From what I learned from the media, there were different measures taken during the Olympic games, like shutting off power plants nearby and building new ones more remotely (which would change the local, but not the total trends, underlining the need for investigations on provincal levels), or reduction of traffic (which would affect the NOx, but not the NOx per fuel). These (and other) different measures and their effect on NOx vs. NOx per fuel trends have to be discussed.*

The measures mentioned by the reviewer for the Olympic Games and other events like World Expo, Youth Olympic Games, EPAC meeting, etc. are mostly of a temporary nature as shown by Mijling et al. (2009). They showed that since most measures were cancelled after the Olympic Games the NO2 levels were back to normal in a couple months.

There have been less permanent regulations for NOx emissions (besides traffic emission regulations) than for SO2, which had for some time a higher priority in China. That is why we have mentioned the NOx regulations in the text, but not made a table for these regulations. However, in the revised version we added a table for the NOx regulations and added some discussion on the traffic regulations and events.

Please note that that we relate NOx emissions to Standard Coal Equivalents (SCE). The reduction of traffic will reduce the NOx emissions but also the NOx per fuel (i.e. NOx per SCE), since NOx is emitted by two different source sectors: (1) traffic and (2) industry including energy production. Of those two source sectors traffic has a higher emission factor (i.e. NO2 per fuel) (see Zhao et al., 2013), thus a shift in source sectors result in different NOx per fuel values. Note that for SO2 this is no issue. To explain this better we have adapted the text (see the next review item).

Zhao, B., Wang, S. X., Liu, H., Xu, J. Y., Fu, K., Klimont, Z., Hao, J. M., He, K. B., Cofala, J., and Amann, M.: NOx emissions in China: historical trends and future perspectives, Atmos. Chem. Phys., 13, 9869-9897, doi:10.5194/acp-13-9869-2013, 2013.

*- NOx per fossil fuel While the SO2 per fuel significantly decreases over the years, in accordance to regulations, the situation is less clear for NOx. The authors state that already 2008 the regulations worked out (explaining the 2009 minimum). But why is NOx per fuel increasing again (by 20%!) in the following years? Have the measures been cancelled? The attempt to explain Fig. 6 by shipping is pure speculation, as it is not supported by any data. The conclusion that NO2 would be 30% higher today without the measures taken is not convincing unless the decline in 2015 compared to the high plateau 2011-2014 is explained; or were all measures concerning NOx just taken in 2015? In any case,*

*the 30% is overestimated as it compares the minimum to the maximum of the timeseries, completely ignoring statistical fluctuations.*

Important for NOx emissions is the growth of the transport sector in China. In the last 10 years the amount of freight transport, expressed in tonnes*km, is approximately doubling every 6 years according to the official statistical information from NBSC (2016). This means that the transport sector is faster growing than the industrial/energy sector, resulting in a gradual shift in the relative shares of the source sectors, which results in gradually higher NOx per fuel in time because of the higher emission factor of traffic (see Zhao et al., 2013). Exception of this gradual growth is the year 2009 because of the following reasons:

- The global economic crisis affected especially the transport sector, which led to a shift in source sector and a reduction of NOx per fossil fuel. See De Ruyter de Wildt et al. (2012)

- The global economic crisis also led to the practice of slow steaming for international ship transport. See Faber et al. (2012) and Boersma et al. (2015).

This explanation has been added to the text. The text has been changed to:

"The year 2009 coincides with the global economic crisis when there was less export of goods from China. This affected especially the transport sector, mostly transport over water, as shown in De Ruyter de Wildt et al. (2012). Faber et al. (2012) and Boersma et al. (2015) showed that the economic crisis also resulted in a significant reduction of the average vessel speed to save fuel used by ship transport. This caused not only a shift in source sectors but in general led to lower NOx per fuel values. This explains the dip in pollution per fuel unit in 2009. After 2012 the gradual increase of NOx per fuel (as a result of the strongly growing transport sector) slowly stops, and the year 2015 shows a sharp decline in $NO_x$ per fossil fuels unit. This can be directly related to the rapidly growing installation of Selective Catalytic Reduction (SCR) equipment at power plants since 2012 and new emission standards for cars as shown by Liu et al. (2016). "

Boersma, K.F., G.C.M. Vinken, J. Tournadre, Ships going slow in reducing their NOx emissions: changes in 2005–2012 ship exhaust inferred from satellite measurements over Europe, Environ. Res. Letters, 10, 7, doi.10.1088/1748-9326/10/7/074007, 2015

De Ruyter de Wildt, M., H. Eskes, K. F. Boersma, The global economic cycle and satellite-derived NO2 trends over shipping lanes, Geophys. Res. Letters, doi:10.1029/2011GL049541, 2012

Faber J, Nelissen D, Hon G, Wang H and Tsimplis M, Regulated slow steaming in maritime transport—an assessment of options, costs and benefits CE Delft (The Netherlands: Delft) (www.cedelft.eu/publicatie/regulated_slow_steaming_in_maritime_transport/1224) p 117 , 2012

The conclusion of a reduction of NOx of 30% as a result of air quality regulations is based on the following:

- Without any air quality regulations we expect a gradual growth in NOx per fossil fuel as a result of the relatively faster growing transport sector that has a higher emission factor. The growth stopped in the year 2012 as a result of the mandatory installation of Selective Catalytic Reduction

**(SCR) equipment at power plants since 2012 and new emission standards for cars (see Table 2). When comparing the year 2015 with the 2012 level we calculate that NOx would be at least (assuming no growth after 2012) of 25% higher without air quality measures. This is different from the earlier 30% because of the new coal numbers we are using in the revised version of the paper. We claim at least 25 % because in this calculation we don't account for the further growth of the transport sector after 2012, while in fact the fast growth of transport still continued.**

**Minor comments:**

*Page 1 Line 13: What does "spatially consistent" mean?*

**We mean that they have about the same quality around the world, but since that is not entirely true, we changed this to "global".**

*Page 1 Line 25: The factor of 3 for SO2 is different from the statement in 3.4, and not supported by the presented data.*

**We changed the text in both cases to "about 2.5 times higher". In 2015 the normalized emission factor is 0.4 compared to around 1 in the period 2003-2007.**

*Page 1 Line 31: "concentrations" should be "column densities"*

**We have made this correction.**

*Page 2 Line 11: Please provide the full name of "He K."*

**We included the full name Kebin He.**

*Page 3 Line 2: Add a reference to Liu et al., 2016: http://www.atmos-chem-phys.net/16/5283/2016/*

**We agree and added the reference.**

*Page 3 Line 12: Please provide some information how and how far the new retrieval improves the quality of SO2.*

**We have added that the new retrieval algorithm "improves the accuracy of the SO$_2$ data for OMI with a factor 2"**

*Page 5 Line 15: NO2 changes during the Olympic games (or during the Shanghai Expo) have been discussed before; please add references.*

**We have added a reference to Mijling et al. (2009), who discuss the effect of the Olympic Games on NOx.**

*Page 6 Line 25: "by definition not sensitive": This is a too strong statement which only holds under the assumption that CHIMERE is doing everything right within the DESCO algorithm.*

**We changed "by definition" into "in general".**

*Page 7 Line 6: There is no "Fig. 5b".*

**We removed the reference.**

---

## Author Response (AR2)

**Response to referee report #1**

We thank Referee #1 for the valuable comments. Below we give a point-to-point response to comments by referee. The referee's comments are given in italic font, while our response is given in bold font.

*My main concern is that the authors fail to put their work into context with the recent publications on the decrease in Chinese emissions. Several of these publications actually present city level trends in pollution and discuss the environmental policy at the city/province level. And several discuss changes in pollutant levels over time. There are also several papers on bottom-up emission inventories that should be reviewed. I recommend that the authors do a thorough search of the literature and discuss the major conclusions of these papers, so as to give credit to the work that has come before. In the process, the importance of your work will be put into perspective.*

**We are aware of the extensive literature on the topic of trends of emissions in China and indeed our discussion on these publications has been deliberately brief. On the other hand the aim of this paper is not to give an review of the existing literature on this topic, but more to focus on literature discussing similar approaches: papers related to satellite-derived trends of concentrations and papers discussing effects of policy making. Note that four of the recent papers on this topic were only published after our original submission of the paper.**

**However, we have expanded the paragraph discussing earlier work in the paper to include more studies on regional or city-scale trends and more recent papers on the decrease in NOx in the last years :**

**"To study the efficiency of the environmental policies, we analysed satellite observations of $SO_2$ and tropospheric $NO_2$ of the last 11 years. $SO_2$ satellite observations over China have been studied earlier by Lee et al. (2010), Li et al (2011), He (2012), Yang et al. (2013), Fioletov et al. (2015), and Krotkov et al. (2015). Satellite observations are very useful for $SO_2$ trend studies, as recently McLinden et al. (2016) showed that bottom-up inventories are underestimating $SO_2$ emissions worldwide with about 0-10%. $NO_2$ satellite observations over China have been evaluated by e.g. Richter et al. (2005), van der A et al. (2006), Zhang et al. (2012), and Krotkov et al. (2015). These studies all showed a strong increase in NO2 over East China. On a city-scale or regional level in China trends are analysed and reported by Gu et al.( 2013), Schneider et al. (2015), and Duncan et al. (2016). Although some cities showed already a decreasing trend, notably in the Pearl river delta. An overall decrease in $NO_2$ concentrations in China is only recently observed by Irie et al. (2016) Liu et al., (2016) and de Foy et al. (2016). To avoid the effects of transport and lifetime of NO2 several authors evaluated NOx emissions instead. Emission estimates of $NO_x$ over China have been analysed by Stavrakou et al. (2008), Kurokawa et al. (2009), and more recently by Mijling et al. (2013) and by Liu et al. (2016a)."**

**Additional references that are used are:**

**Duncan B N, Lamsal L N, Thompson A M, Yoshida Y, Lu Z, Streets D G, Hurwitz M, M and Pickering K E, 2016, A space-based, high-resolution view of notable changes in urban NOx pollution around the world (2005–2014),J. Geophys. Res. 121, 976–96**

**Gu D, Wang Y, Smeltzer C and Liu Z, 2013, Reduction in NOx emission trends over China: regional and seasonal variations, Environ. Sci. Technol., 47, 12912–9**

**Schneider, P., W.A. Lahoz and R.J. van der A, Recent Satellite-based Trends of Tropospheric Nitrogen Dioxide over Large Urban Agglomerations Worldwide, Atm. Chem. Phys., 2015, 15, 1205-1220, doi:10.5194/acp-15-1205-2015.**

**B. de Foy, Z. Lu and D. G. Streets, Satellite NO2 retrievals suggest China has exceeded its NOx reduction goals from the twelfth Five-Year Plan, Scientific Reports 6, Article number: 35912 (2016) doi:10.1038/srep35912**

**Response to referee report #2**

We thank Referee #2 for the valuable comments. Below we give a point-to-point response to comments by referee. The referee's comments are given in italic font, while our response is given in regular font.

5 *The authors have modified the original manuscript, but several of the requested "major revisions" have not been resolved. I cannot recommend publication before the following issues have been adressed adequately:*

*1. My main concern about the study is that the authors are quite bold with there statement that Chinese NO2 levels would be higher by 30% (now 25%) without regulations.*

*This is quite speculative and not supported by the shown data: on the one hand, the authors claim that regulations were*
10 *increasingly taken over the last years, while on the other hand the 25% reduction is just estimated based on the value in 2015. So if the authors would have had written their study one year earlier, based on data for 2005-2014, they would have had to conclude that the many regulations taken so far are useless!*

*So unless the authors provide a good (and quantitative) explanation on why the NOx per fossil fuel ratio kept almost constant in 2010-2014, despite regulations, they cannot claim that the oberved reduction in 2015 allows for a direct*
15 *interpretation as "this is the accurately quantified benefit of regulation efforts".*

**We claim that the reduction is at least 25 % (actually 27%) but we are also considering errors in the fossil fuel consumption numbers. This is a cautious estimate based on the years 2011 (the last year that is still not much affected by SCR installations) and the last year 2015. Without regulations we expect a continuous steady growth after 2012 in the NOx per fossil fuel consumption, because the percentage of the transport sector is increasing. (The transport**
20 **sector is emitting much more NOx per fossil fuel unit than the energy or industrial sectors.) This conclusion is based on the statistics of the NBSC (2015) and Wu et al. (2017), which shows that the transport is growing with a factor two every 5 years. This is much faster than any other sector is growing.**

**We have adapted the text to clarify this point at several places in the text at the end of section 3.**

**Important point in our conclusion is that regulations for NOx, unlike for SO2, were effective not earlier than 2012**
25 **and especially a big effort is made in the last two years as shown in Liu et al., (2016). We see the growth in NOx per fossil fuel stop and slowly decrease after 2012.**

**Interesting to see is that the decrease in NOx emission seems to be behind the actual installation of SCR technology. We saw something similar for SO2 where not the actual implementation of filtering was important but the control of the use of these filters a few years later made the change (see Xu, 2011 ). But if this is also the case for NOx filters is**
30 **pure speculation at this point.**

*I already made this point in my first review. The reply given by the authors and their modifications in the text expand the discussion of (a) the changing role of transport and shipping, and (b) the effectiveness of SCR equipment.*
35 *However, (a) is provided qualitatively (which is fine for undestanding the general pattern, but far too vague to assess the robustness of the 25% reduction estimate),*

**Please note that it is very hard to get reliable statistics (or numbers at all) related to emissions in China. The exact numbers for transport and shipping are not available to us, however information on the relative growth of traffic is given in NBSC and Wu et al. (2017) and amount to a doubling every 5-6 years. However, this is not enough for**
40 **quantitatively analysis, which is the reason that we say the reduction is "at least 25 %", while due to higher contribution of traffic this should even be higher. We have given some clarification on this point in the text at the end of section 3.**

*while (b) refers to a submitted paper, not accessible yet, and now suddenly claims SCR equipment playing a key role (something not mentioned at all in the first version nor in the introduction of the second version).*

**Meanwhile, the paper has been published on 24 October in Environmental Research Letters. We hope that this take away the objection of the referee on this point. We have added in the introduction:**

5 **"According to Liu et al. (2016) Selective Catalytic Reduction (SCR) equipment was installed in this period and growing from a penetration of about 18% in 2011 until 86% in 2015. SCR equipment in power plants are expected to reduce the emissions of the power plant with at least 70% (ICAC, 2009). The SCR installation is the most significant measure taken to reduce the $NO_x$ emissions and it largely coincides with the peak year of observed $NO_2$ concentrations (Liu et al., 2016)."**

10 **"To our knowledge no regulations for ship emissions have been announced. "**

*In addition I would like to point out that if you have to consider changes of the relative role of traffic (without having a specific number for this), the concept of the simple NOx/fuel ratio is meaningless.*

15 **Originally this concept was designed for SO2, while for NOx it is somewhat more difficult to interpret. However, we certainly do not see it as meaningless since based on the limited information we have we can still conclude that the environmental regulations, especially regarding SCR, are very effective recently.**

**Note that we really see a strong decrease in observed NO2 concentrations over China. By dividing it by fossil fuel we put it in perspective: in this way we see a trend in emission factor instead of emissions themselves. Whether this is**
20 **caused by shifts in the importance of certain sectors or by air quality regulations has been verified by additional information.**

*So the authors have to actually explain the years 2010-2014 as well as the year 2015 with data for (ship) traffic, SCR, etc.*
25 *(including some error estimate of the NOx/fuel ratio, which is affected by uncertainties in both NOx emissions and fossil fuel inventory), or have to weaken their statements on NOx reduction.*

**We have included explanations for the growth and reduction of NOx per fossil fuel in the entire period of our research. The growth can be explained by the growing transport sector, the reduction in 2009 is a result of less transport due to the global economic recession and the last period of strong decrease of NOx emission is due to**
30 **regulations for power plants (installing SCR) and on-road vehicle emission regulations. This has been explained in the text with related references. In addition we added the following remarks:**

**"After 2009 the $NO_x$ per fossil fuel is slowly increasing because the transport sector is growing faster than the energy sector and has a higher emission factor. Statistics of the NBSC (2015) show that the transport is growing with a factor two every 5-6 years (Wu et al., 2017)."**

35 **"For $NO_x$ per fossil fuel we were expecting a gradual growth after 2012 because of the continuing relative growth of the transport sector. Keeping this in mind we compare the years 2015 with 2012 and conclude that without air quality regulations the $NO_2$ concentrations would be at least 25% higher in China today"**

**The uncertainty in NOx emissions is reduced to a negligible level as explained in the text due to the averaging over the province area and averaging over a time period of 12 months. The uncertainty of the fossil fuel numbers is much**
40 **higher but difficult to quantify. Papers on this topic are analysing earlier versions of the fossil fuel inventory. They show that the uncertainty varies over time and per province. Since the uncertainty of our fossil fuel numbers is unknown, we decided not to discuss this in the paper.**

**Wu, Y., S. Zhang, J. Hao, H. Liu, X. Wu, J. Hu, M. P. Walsh, T. J. Wallington, K. M. Zhang, S. Stevanovic, On-road vehicle emissions and their control in China: A review and outlook, Science of The Total Environment, 574, 332-349, doi:10.1016/j.scitotenv.2016.09.040, 2017**

*2. The topic of the paper is regulation, but the reader learns almost nothing about it.*

*In my first review, I have asked to "discuss the different possibilities in general, and the taken measures in detail". While the authors have addressed the second point in a new table, the general possibilities/techniques for reducing NOx emissions for cars, PPs etc. are not discussed in the introduction. This has to be added (including SCR!) with appropriate references.*

10 **We have added in the introduction of the revised version more details on the air quality regulations concerning the SCR equipment installations and the new car standards. Also some discussion on the expected NOx reductions is included. This resulted in the following new text:**

**" (…..) According to Liu et al. (2016) Selective Catalytic Reduction (SCR) equipment was installed in this period and growing from a penetration of about 18% in 2011 until 86% in 2015. SCR equipment in power plants are expected to**

15 **reduce the emissions of the power plant with at least 70% (ICAC, 2009). The SCR installation is the most significant measure taken to reduce the $NO_x$ emissions and it largely coincides with the peak year of observed $NO_2$ concentrations (Liu et al., 2016). At the same time China has implemented several new national emission standards for cars during the time period of our study (see Table 2). The change from China 3 to China 4 standard for cars in the period 2011-2015 reduces the maximum allowed amount of $NO_x$ emissions for on-road vehicles with 50% (Wu et**

20 **al., 2017). More strict regulations for on-road vehicles (e.g. ban on older polluting cars) have been introduced in China on a provincial or even city level, e.g. in Beijing, rather than nationwide. To our knowledge no regulations for ship emissions have been announced.(….)"**

*3. Finally, I have asked for analysis on province level, as well as the other reviewer. While the authors now provide*

25 *provincial data, they do not actually provide provincial analysis. This, however, would provide a good opportunity to shed light on the discussion of the effectiveness of regulations. So, how does the NOx per fuel ratio look like for different provinces? Is it different for coastal provinces (with significant ship emissions)? Can you relate the temporal patterns per province to SCR equipment? Are the temporal patterns for Shanghai Expo/Olympic games/APEC matching our expectations?*

30 *Such an in-detail analysis on provincal level would reveal how robust the presented ratio actually is.*

**Unfortunately the provincial data on coal and oil consumption are not reliable enough to do these analyses for all provinces. Most uncertainties occur in the numbers related to oil, although also coal consumption numbers are not always correct. However, for the coal-consumption dominated provinces an analysis is possible, and that is what we added to the paper with the following discussion:**

35 **"On a provincial scale we can in principle do similar analyses, but unfortunately the provincial energy consumption related to coal and oil have a very high uncertainty due to inconsistencies in interprovincial imports and exports (Hong et al., 2016). We see this reflected in a high variability in the annual provincial data and sometimes missing data. Especially for the oil consumption the data have high uncertainties (Guan et al., 2011, Hong et al., 2016). Therefore, we have only analysed the 5 provinces with dominating coal consumption as shown in Figure 8. In this**

40 **graph we excluded Guizhou province because of its difficult to interpret coal consumption in 2011 as a result of large power shortages (NBSC, 2015, Xinhua, 2011). For SO2 per fossil fuel unit we see that all provinces follow the national trend. For NOx per fossil fuel we see more variation per province depending on the role of transport. Most of these by coal consumption dominated provinces start their decreasing trend from 2011 reflecting the national program on**

SCR installations starting that same year. It is interesting to see that the commissioning of new power plants in 2011 causes a strong increase in both SO2 and NOx in the province Ningxia (see Figure 4 and 5a). However, when compensating for fossil fuel usage, one can see that the same national air quality regulations are applied here, as the trend in Figure 8 shows the same pattern as for other provinces. This show the strength of the presented method to assess the efficiency of air pollution regulations."

The SCR and other air quality measures are clearly visible in the added Figure 8.

We are not focusing too much on a discussion on the temporary air quality regulations for certain cities since these measures are lasting for a few days or a couple of months at most and it is not the purpose of this paper to analyse these events. We have (co-)authored several other papers on this topic. Only for small provinces and big events like the Olympic Games in Beijing and World Expo in Shanghai it enters our present results. Therefore we have only added the following sentence:

"Events like the Olympic Games in Beijing in 2008 and the World Expo in Shanghai in 2010, when temporary strict air quality regulations have been enforced, can be recognised in this Table as years with significant lower emissions for these provinces."

*Additional comments:*

*- Section 3.3: "Total NOx emissions in East China reached their peak levels in 2012, and have been decreasing since".*
*From Figure 5, I would rather claim that overall emissions have stayed constant for 2012-2014, and decreased in 2015.*

We have changed this sentence to "Total $NO_x$ emissions in East China reached their peak levels in 2012, and have stopped increasing since this year"

*- Table 2: Please include a column on the expected NOx reduction.*

We have added in the text in the introduction some discussion on the expected NOx reductions (see answer to point 2)

The next pages contain the marked-up manuscript

[revised manuscript text omitted]

---

## Author Response (AR3)

**Response to the editor report**

For clarity we have repeated the editor report below in italic font with our response following in bold font. The manuscript with track changes follows on the following pages.

5 *Co-Editor Decision: Reconsider after minor revisions (Editor review) (29 Dec 2016) by Gregory Frost*

*Comments to the Author:*

*I would like the authors to expend a bit more effort addressing the comments of Referee #1, which I feel are justified and appropriate. Please be more explicit about what new insights you have gained with your current analysis, and tell us why these insights extend our knowledge beyond what has already been learned from the large body of work on emissions*

10 *changes in China.*

*The reporting of previous work in the Introduction is fine. But that doesn't tell us why your analysis presents an advance over what we already knew.*

*This should not require much additional text. I'm looking for a few sentences or a paragraph in the Discussion.*

15 **We probably have misunderstood the previous comments of Referee #1. To resolve this issue we have added the following paragraph to the Discussion as the editor suggested:**

[revised manuscript text omitted]